# Vitruvion: A Generative Model of Parametric CAD Sketches

**Ari Seff**[1], **Wenda Zhou**[2,3], **Nick Richardson**[1], **& Ryan P. Adams**[1]
[1]Princeton University, [2]New York University, [3]Flatiron Institute
{aseff,njkrichardson, rpa}@princeton.edu
{wz2247}@nyu.edu

## Abstract

Parametric computer-aided design (CAD) tools are the predominant way that engineers specify physical structures, from bicycle pedals to airplanes to printed circuit boards. The key characteristic of parametric CAD is that design intent is encoded not only via geometric primitives, but also by parameterized constraints between the elements. This relational specification can be viewed as the construction of a constraint program, allowing edits to coherently propagate to other parts of the design. Machine learning offers the intriguing possibility of accelerating the design process via generative modeling of these structures, enabling new tools such as autocompletion, constraint inference, and conditional synthesis. In this work, we present such an approach to generative modeling of parametric CAD sketches, which constitute the basic computational building blocks of modern mechanical design. Our model, trained on real-world designs from the SketchGraphs dataset, autoregressively synthesizes sketches as sequences of primitives, with initial coordinates, and constraints that reference back to the sampled primitives. As samples from the model match the constraint graph representation used in standard CAD software, they may be directly imported, solved, and edited according to downstream design tasks. In addition, we condition the model on various contexts, including partial sketches (primers) and images of hand-drawn sketches. Evaluation of the proposed approach demonstrates its ability to synthesize realistic CAD sketches and its potential to aid the mechanical design workflow.

## 1 Introduction

Parametric computer-aided design (CAD) tools are at the core of the design process in mechanical engineering, aerospace, architecture, and many other disciplines. These tools enable the specification of two- and three-dimensional structures in a way that empowers the user to explore parameterized variations on their designs, while also providing a structured representation for downstream tasks such as simulation and manufacturing. In the context of CAD, *parametric* refers to the dominant paradigm for professionals, in which the software tool essentially provides a graphical interface to the specification of a *constraint program* which can then be solved to provide a geometric configuration. This implicit programmatic workflow means that modifications to the design—alteration of dimensions, angles, etc.—propagate across the construction, even if it is an assembly with many hundreds or thousands of components.

At an operational level, parametric CAD starts with the specification of two-dimensional geometric representations, universally referred to as "sketches" in the CAD community (Shah, 1998; Camba et al., 2016; Choi et al., 2002). A sketch consists of a collection of geometric primitives (e.g., lines, circles), along with a set of *constraints* that relate these primitives to one another. Examples of sketch constraints include parallelism, tangency, coincidence, and rotational symmetry. Constraints are central to the parametric CAD process as they reflect abstract design intent on the part of the

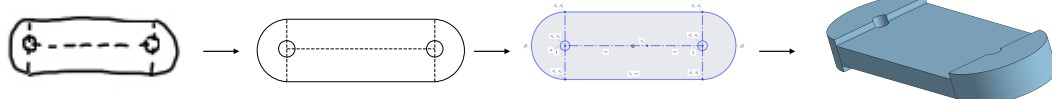

Figure 1: CAD sketch generation conditioned on a hand drawing. Our model first conditions on a raster image of a hand-drawn sketch and generates sketch primitives. Given the primitives, the model generates constraints, and the resulting parametric sketch is imported and edited in CAD software, ultimately producing a 3D model.

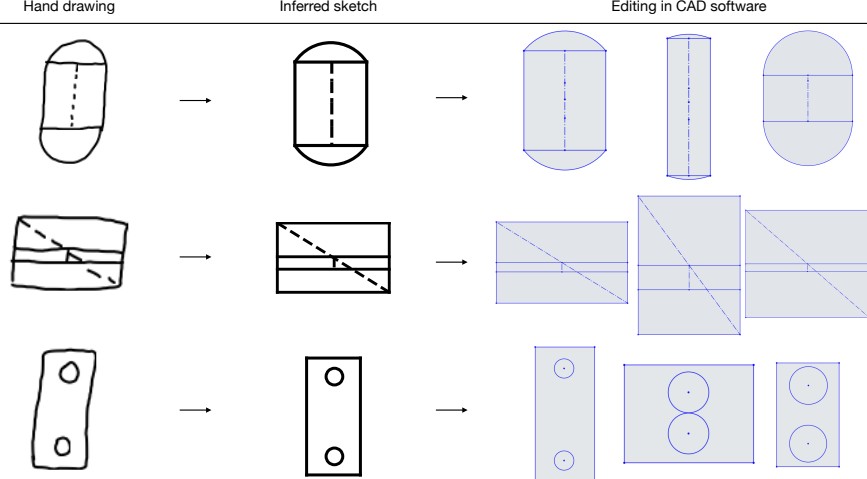

Figure 2: Editing inferred sketches. We take raster images of hand-drawn sketches (left), infer the intended sketches using our image-conditional primitive model and constraint model (center), then solve the resultant sketches in CAD software, illustrating edit propagation arising from the constraints (right).

user, making it possible for numerical modifications to coherently propagate into other parts of the design. Figure 2, which demonstrates a capability made possible by our approach, illustrates how constraints are used to express design intent: the specific dimensions of the final model (shown on the right) can be modified while preserving the geometric relationships between the primitives. These constraints are the basis of our view of parametric CAD as the specification of a program; indeed the actual file formats themselves are indistinguishable from conventional computer code.

This design paradigm, while powerful, is often a challenging and tedious process. An engineer may execute similar sets of operations many dozens of times per design. Furthermore, general motifs may be repeated across related parts, resulting in duplicated effort. Learning to accurately predict such patterns has the potential to mitigate the inefficiencies involved in repetitive manual design. In addition, engineers often begin visualizing a design by roughly drawing it by hand (Fig. 1). The automatic and reliable conversion of such hand drawings, or similarly noisy inputs such as 3D scans, to parametric, editable models remains a highly sought feature.

In this work, we introduce Vitruvion, a generative model trained to synthesize coherent CAD sketches by autoregressively sampling geometric primitives and constraints (Fig. 3). The model employs self-attention (Vaswani et al., 2017) to flexibly propagate information about the current state of a sketch to a next-step prediction module. This next-step prediction scheme is iterated until the selection of a stop token signals a completed sample. The resultant geometric constraint graph is then handed to a solver that identifies the final configuration of the sketch primitives. By generating sketches via constraint graphs, we allow for automatic edit propagation in standard CAD software. Constraint supervision for the model is provided by the SketchGraphs dataset Seff et al. (2020), which pairs millions of real-world sketches with their ground truth geometric constraint graphs.

In addition to evaluating the model via log-likelihood and distributional statistics, we demonstrate the model's ability to aid in three application scenarios when conditioned on specific kinds of context. In **autocomplete**, we first prime the model with an incomplete sketch (e.g., with several primitives missing) and query for plausible completion(s). In **autoconstrain**, the model is conditioned on a set of primitives (subject to position noise), and attempts to infer the intended constraints. We also explore **image-conditional synthesis**, where the model is first exposed to a raster image of a hand-drawn sketch. In this case, the model is tasked with inferring both the primitives and constraints corresponding to the sketch depicted in the image. Overall, we find the proposed model is capable of synthesizing realistic CAD sketches and exhibits potential to aid the mechanical design workflow.

## 2 RELATED WORK

When discussing related work, is important to note that in computer-aided design, the word "sketch" is a term of art that refers to this combination of primitives and constraints. In particular, it should

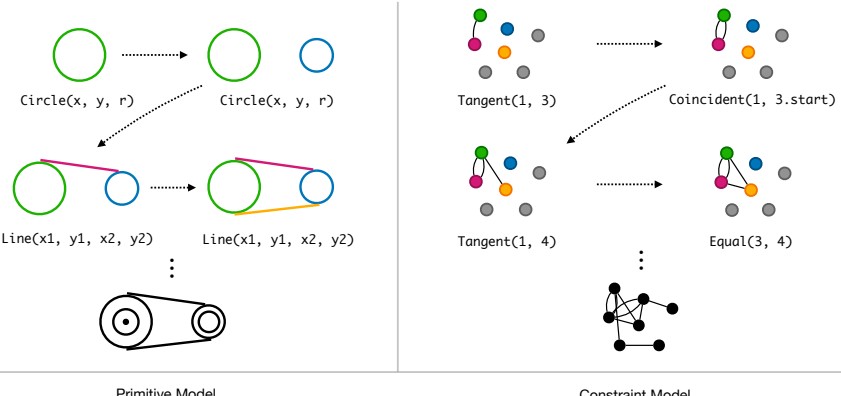

Figure 3: We factorize CAD sketch synthesis into two sequence modeling tasks: primitive generation (left) and constraint generation (right). The constraint model conditions on the present primitives and outputs a sequence of constraints, serving as edges in the corresponding geometric constraint graph. A separate solver (standard in CAD software) is used to adjust the primitive coordinates, if necessary.

not be confused with the colloquial use of the word "sketch" as has sometimes appeared in the machine learning literature, as in Ha & Eck (2018) discussed below.

**CAD sketch generation** The SketchGraphs dataset (Seff et al., 2020) was recently introduced to target the underlying relational structure and construction operations found in mechanical design software. Specifically, the dataset provides ground truth geometric constraint graphs, and thus the original design steps, for millions of real-world CAD sketches created in Onshape.[1] Initial models are trained in Seff et al. (2020) for constraint graph generation using message-passing networks, but without any learned attributes for the positions of the primitives; the sketch configuration is determined via constraints alone, limiting the sophistication of output samples.

Following SketchGraphs, several works concurrent to ours explore generative modeling of CAD sketches utilizing transformer-based architectures (Willis et al., 2021; Ganin et al., 2021; Para et al., 2021; Wu et al., 2021). In Willis et al. (2021), modeling is limited to primitives, without any learned specification of geometric constraints; Vitruvion, however, enables edit propagation in standard CAD software by outputting constraints. Similar to our approach, both primitives and constraints are modeled in Ganin et al. (2021); Para et al. (2021). In contrast to our work, these do not condition on hand-drawn sketches nor consider noisy inputs when training on the constraint inference task (which proves crucial for generalization in Section 4.4). Unlike the above works that solely target 2D sketches (including ours), Wu et al. (2021) attempts to model sequences of both 2D and 3D CAD commands, but, similarly to (Willis et al., 2021), does not output sketch constraints.

**Geometric program synthesis** CAD sketch generation may be viewed via the broader lens of geometric program synthesis. This comprises a practical subset of program synthesis generally concerned with inferring a set of instructions (a program) for reconstructing geometry from some input specification. Recent examples include inferring TikZ drawing commands from images (Ellis et al., 2018) and constructive solid geometry programs from voxel representations (Sharma et al., 2018). Related to these efforts is the highly-sought ability to accurately reverse engineer a mechanical part given underspecified and noisy observations of it (e.g., scans, photos, or drawings). One of the conditional generation settings we explore in this work is training a version of our model that generates parametric sketches when given a raster hand drawing. Note that previous work on geometric program synthesis have not generally considered constraints, which are critical to the CAD application.

**Vector graphics generation** Generative models have been successfully applied to vector graphics in recent years. Sketch-RNN (Ha & Eck, 2018) is trained on sequences of human drawing strokes to learn to produce vector drawings of generic categories (dog, bus, etc.). Note the common meaning of the word *sketch* differs slightly from its usage as a term of art in mechanical design (as we use it in this work). DeepSVG (Carlier et al., 2020) models SVG icons, demonstrating flexible editing with latent space interpolation. Less recent are tracing programs, which can convert raster drawings to vector graphics and have been widely available for decades (e.g., Selinger (2003)). None of this previous work considers constraints as a mechanism for specifying design intent as performed in

---

[1]https://www.onshape.com

CAD. The image-conditional version of Vitruvion also takes as input a raster image (hand-drawn) and can output vector graphics. Unlike tracing, which attempts to faithfully reproduce an input image as closely as possible, our model attempts to infer the sequence of geometric primitives and constraints intended by the user, learning to ignore noise in the input.

**Graph-structured modeling** This work is also related to modeling of graph structures, via our use of geometric constraint graphs. Graphs constitute a natural representation of relational structure across a variety of domains, including chemistry, social networks, and robotics. In recent years, message-passing networks (Gilmer et al., 2017; Duvenaud et al., 2015; Kipf & Welling, 2017), building off of Scarselli et al. (2009), have been extensively applied to generative modeling of graph-structured data, such as molecular graphs (Jin et al., 2018; Liu et al., 2018). These networks propagate information along edges, learning node- or graph-level representations.

While message passing between adjacent nodes can be a useful inductive bias, it can be an impediment to effective learning when communication between distant nodes is necessary, e.g., in sparse graphs (Kurin et al., 2021). The constraint graphs in our work are sparse, as the constraints (edges) grow at most linearly in the number of primitives (nodes). Recent work leverages the flexible self-attention mechanism of transformers (Vaswani et al., 2017) to generate graph-structured data without the limitations of fixed, neighbor-to-neighbor communication paths, such as for mesh generation (Nash et al., 2020). Our work similarly bypasses edge-specific message passing, using self-attention to train a generative model of geometric constraint graphs representing sketches.

## 3    METHOD

We aim to model a distribution over parametric CAD sketches. Each sketch is comprised of both a sequence of primitives, $\mathcal{P}$, and a sequence of constraints, $\mathcal{C}$. We note that these are sequences, as opposed to unordered sets, due to the ground truth orderings included in the data format (Seff et al., 2020). Primitives and constraints are represented similarly in that both have a categorical type (e.g., Line, Arc, Coincidence) as well as a set of additional parameters which determine their placement/behavior in a sketch. Each constraint includes *reference* parameters indicated precisely which primitive(s) (or components thereof) must adhere to it. Rather than serving as static geometry, CAD sketches are intended to be dynamic—automatically updating in response to altered design parameters. Likewise, equipping our model with the ability to explicitly constrain generated primitives serves to ensure the preservation of various geometric relationships during downstream editing.

We divide the sketch generation task into three subtasks: primitive generation, constraint generation, and constraint solving. In our approach, the first two tasks utilize learned models, while the last step may be conducted by any standard geometric constraint solver.[2] We find that independently trained models for primitives and constraints allows for a simpler implementation. This setup is similar to that of Nash et al. (2020), where distinct vertex and face models are trained for mesh generation.

The overall sketch generation proceeds as:

$$p_\theta(\mathcal{P}, \mathcal{C} \mid \text{ctx}) = p_\theta(\mathcal{C} \mid \mathcal{P})p_\theta(\mathcal{P} \mid \text{ctx}) \qquad \mathcal{S} = \text{solve}(\mathcal{P}, \mathcal{C}) \qquad (1)$$

where:

- $\mathcal{P}$ is a sequence of primitives, including parameters specifying their positions;
- $\mathcal{C}$ is a sequence of constraints, imposing mandatory relationships between primitives;
- $\mathcal{S}$ is the resulting sketch, formed by invoking a solve routine on the $(\mathcal{P}, \mathcal{C})$ pair; and
- ctx is an optional context for conditioning, such as an image or prefix (primer).

This factorization assumes $\mathcal{C}$ is conditionally independent of the context given $\mathcal{P}$. For example, in an image-conditional setting, access to the raster representation of a sketch is assumed to be superfluous for the constraint model given accurate identification of the portrayed primitives and their respective positions in the sketch. Unconditional samples may be generated by providing a null context.

### 3.1    PRIMITIVE MODEL

The primitive model is tasked with autoregressive generation of a sequence of geometric primitives. For each sketch, we factorize the distribution as a product of successive conditionals,

$$p_\theta(\mathcal{P} \mid \text{ctx}) = \prod_{i=1}^{N_\mathcal{P}} p_\theta(\mathcal{P}_i \mid \{\mathcal{P}_j\}_{j<i}, \text{ctx}), \qquad (2)$$

---

[2]We use D-Cubed (Siemens PLM) for geometric constraint solving via Onshape's public API.

where $N_{\mathcal{P}}$ is the number of primitives in the sequence. Each primitive is represented by a tuple of tokens indicating its type and its parameters. The primitive type may be one of four possible shapes (here either arc, circle, line, or point), and the associated parameters include both continuous positional coordinates and an `isConstruction` Boolean.[3] We use *sequence* to emphasize that our model treats sketches as constructions arising from a step-by-step design process. The model is trained in order to maximize the log-likelihood of $\theta$ with respect to the observed token sequences.

**Ordering.** Abstractly, the mapping from constructions sequences to final sketch geometries is clearly non-injective; it is typical for many construction sequences to result in the same final geometry. Neverthe­less, empirically, we know that the design routes employed by engineers often admit

| Arc | Circle | Line | Point |
|---|---|---|---|
| $(x_1, y_1, x_{\mathrm{mid}}, y_{\mathrm{mid}}, x_2, y_2)$ | $(x, y, r)$ | $(x_1, y_1, x_2, y_2)$ | $(x, y)$ |

Table 1: We convert the native Onshape numerical parame­ters to the above forms for modeling. Subscripts of 1, mid, and 2 indicate start, mid, and endpoints, respectively.

evident patterns, e.g., greater involvement of earlier "anchor" primitives in constraints (Seff et al., 2020). In our training data from SketchGraphs, the order of the design steps taken by the original sketch creators is preserved. Thus, we expose the model to this ordering, training our model to autoregressively predict subsequent design steps conditioned on prefixes. By following an explicit representation of this ordering, our model is amenable to applications in which conditioning on a prefix of design steps is required, e.g., autocompleting a sketch. This approach also constrains the target distribution such that the model is not required to distribute its capacity among all orderings.

**Parameterization.** SketchGraphs uses the parameterization provided by Onshape. In order to ac­commodate the interface to the Onshape constraint solver, these are often *over-parameterizations*, e.g. line segments are defined by six continuous parameters rather than four. We compress the orig­inal parameterizations into minimal canonical parameterizations for modeling (Table 1) and inflate them back to the original encoding prior to constraint solving.

**Normalization.** The sketches in SketchGraphs exhibit widely varying scales in physical space, from millimeters to meters. In addition, because Onshape users are free to place sketches anywhere in the coordinate plane, they are often not centered with respect to the origin in the coordinate space. To reduce modeling ambiguity, we modify the primitive coordinates such that each sketch's square bounding box has a width of one meter and is centered at the origin.

**Quantization.** The sketch primitives are inherently described by continuous parameters such as $x, y$ Cartesian coordinates, radii, etc. As is now a common approach in various continuous domains, we quantize these continuous variables and treat them as discrete. This allows for more flexible, yet tractable, modeling of arbitrarily shaped distributions (van den Oord et al., 2016). Following the sketch normalization procedure described in Section 3.1, we apply 6-bit uniform quantization to the primitive parameters. Lossy quantization is tolerable as the constraint model and solver are ultimately responsible for the final sketch configuration.

**Tokenization.** Each parameter (whether categorical or numeric) in the input primitive sequence is represented using a three-tuple comprised of a value, ID, and position token. Value tokens play a dual role. First, a small portion of the range for value tokens is reserved to enumerate the primitive types. Second, the remainder of the range is treated separately as a set of (binned) values to specify the numerical value of some parameter, for example, the radius of a circle primitive. ID tokens specify the type of parameter being described by each value token. Position tokens specify the ordered index of the primitive that the given element belongs to. See Section C for further details on the tokenization procedure.

**Embeddings.** We use learned embeddings for all tokens. That is, each value, ID, and position token associated with a parameter is independently embedded into a fixed-length vector. Then, we compute an element-wise sum across each of these three token embedding vectors to produce a single, fixed-length embedding to represent the parameter embedding. The sequence of parameter embeddings are then fed as input to the decoder described below.

**Architecture.** The unconditional primitive model is based on a decoder-only transformer (Vaswani et al., 2017) operating on the flattened primitive sequences. To respect the temporal dependencies of the sequence during parallel training, the transformer decoder leverages standard triangular masking

---

[3]*Construction* or *virtual* geometry is employed in CAD software to aid in applying constraints to regular geometry. `isConstruction` specifies whether the given primitive is to be physically realized (when false) or simply serve as a reference for constraints (when true).

during self-attention. The last layer of the transformer outputs a final representation for each token that then passes through a linear layer to produce a logit for each possible value token. Taking a softmax then gives the categorical distribution for the next-token prediction task. We discuss conditional variants in Section 3.3. See the appendix for further details.

## 3.2 CONSTRAINT MODEL

Given a sequence of primitives, the constraint model is tasked with autoregressive generation of a sequence of constraints. As with the primitive model, we factorize the constraint model as a product of successive conditionals,

$$p_\theta(\mathcal{C} \mid \mathcal{P}) = \prod_{i=1}^{N_\mathcal{C}} p_\theta(\mathcal{C}_i \mid \{\mathcal{C}_j\}_{j<i}, \mathcal{P}), \qquad (3)$$

where $N_\mathcal{C}$ is the number of constraints. Each constraint is represented by a tuple of tokens indicating its type and parameters. Like the primitive model, the constraint model is trained in order to maximize the log-likelihood of $\theta$ with respect to the observed token sequences.

Here, we focus our constraint modeling on handling all categorical constraints containing one or two reference parameters. This omits constraints with numerical parameters (such as scale constraints) as well as "hyperedge" constraints that have more than two references.

**Ordering.** We canonicalize the order of constraints according to the order of the primitives they act upon, similar to Seff et al. (2020). That is, constraints are sorted according to their latest member primitive. For each constraint type, we arbitrarily canonicalize the ordering of its tuple of parameters. Constraint tuples specify the type of constraint and its reference parameters.

**Tokenization.** The constraint model employs a similar tokenization scheme to the primitive model in order to obtain a standardized sequence of integers as input. For each parameter, a triple of value, ID, and position tokens indicate the parameter's value, what type of parameter it is, and which ordered constraint it belongs to, respectively.

**Architecture.** The constraint model employs an encoder-decoder transformer architecture which allows it to condition on the input sequence of primitives. A challenge with the constraint model is that it must be able to specify references to the primitives that each constraint acts upon, but the number of primitives varies among sketches. To that end, we utilize variable-length softmax distributions (via pointer networks (Vinyals et al., 2015)) over the primitives in the sketch to specify the constraint references. This is similar to (Nash et al., 2020), but distinct in that our constraint model is equipped to reference hierarchical entities (i.e., primitives or sub-primitives).

**Embeddings.** Two embedding schemes are required for the constraint model. Both the input primitive token sequence as well as the target output sequence (constraint tokens) must be embedded. The input primitive tokens are embedded identically to the standalone primitive model, using lookup tables. A transformer encoder then produces a sequence of final representations for the primitive tokens. These vectors serve a dual purpose: conditioning the constraint model and embedding references in the constraint sequence. The architecture of the encoder is similar to that of the primitive model, except it does not have the output linear layer and softmax. From the tuple of representations for each primitive, we extract reference embeddings for both the primitive as a whole (e.g., a line segment) and each of its sub-primitives (e.g., a line endpoint) that may be involved in constraints.

**Noise injection.** The constraint model must account for potentially imperfect generations from the preceding primitive model. Accordingly, the constraint model is conditioned on primitives whose parameters are subjected to independent Gaussian noise during training.

## 3.3 CONTEXT CONDITIONING

Our model may be optionally conditioned on a context. As described below Eq. (1), we directly expose the primitive model to the context while the constraint model is only implicitly conditioned on it via the primitive sequence. Here, we consider two specific cases of application-relevant context: primers and images of hand-drawn sketches.

**Primer-conditional generation.** Here, a primer is a sequence of primitives representing the prefix of an incomplete sketch. Conditioning on a primer consists of presenting the corresponding prefix to a trained primitive model; the remainder of the primitives are sampled from the model until a stop token is sampled, indicating termination of the sequence. This emulates a component of an "auto-complete" application, where CAD software interactively suggests next construction operations.

**Image-conditional generation.** In the image-conditional setting, we are interested in accurately recovering a parametric sketch from a raster image of a hand-drawn sketch. This setup is inspired by the fact that engineers often draw a design on paper or whiteboard from various orthogonal views prior to investing time building it in a CAD program. Accurate inference of parametric CAD models from images or scans remains a highly-sought feature in CAD software, with the potential to dramatically reduce the effort of translating from rough paper scribble to CAD model.

When conditioning on an image, the primitive model is augmented with an image encoder. We leverage an architecture similar to a vision transformer (Dosovitskiy et al., 2021), first linearly projecting flattened local image patches to a sequence of initial embeddings, then using a transformer encoder to produce a learned sequence of context embeddings. The primitive decoder cross-attends into the image context embedding. The overall model is trained to maximize the probability of the ground truth primitives portrayed in the image; we condition on a sequence of patch embeddings. This is similar to the image-conditioning route taken in Nash et al. (2020) for mesh generation, which also uses a sequence of context embeddings produced by residual networks (He et al., 2016).

**Hand-drawn simulation.** We aim to enable generalization of the image-conditional model described above to hand-drawn sketches, which can potentially support a much wider array of applications than only conditioning on perfect renderings. This requires a noise model to emulate the distortions introduced by imprecise hand drawing, as compared to the precise rendering of a parametric sketch using software. Two reasonable noise injection targets are the parameters underlying sketch primitives and the raster rendering procedure. Our noise model takes a hybrid approach, subjecting sketch primitives to random translations/rotations in sketch space, and augmenting the raster rendering with a Gaussian process model. A full description is provided in Section B.

## 4 EXPERIMENTS

We evaluate several versions of our model in both primitive generation and constraint generation settings. Quantitative assessment is conducted by measuring negative log-likelihood (NLL) and predictive accuracy on a held-out test set as well as via distributional statistics of generated sketches. We also examine the model's performance on conditional synthesis tasks.

### 4.1 TRAINING DATASET

Our models are trained on the SketchGraphs dataset (Seff et al., 2020), which consists of several million CAD sketches collected from Onshape. We use the filtered version, which is restricted to sketches comprised of the four most common primitives (arcs, circles, lines, and points), a maximum of 16 primitives per sketch, and a non-empty set of constraints. We further filter out any sketches with fewer than six primitives to eliminate most trivial sketches (e.g., a simple rectangle). We randomly divide the filtered collection of sketches into a 92.5% training, 2.5% validation, and 5% testing partition. Training is performed on a server with four Nvidia V100 GPUs, and our models take between three and six hours to train. See Section E.2 for full details of the training procedure.

**Deduplication.** As components of larger parts and assemblies, CAD sketches may sometimes be reused across similar designs. Onshape's native document-copying functionality supports this. In addition, similar sketches may result from users of Onshape following tutorials, importing files from popular CAD collections, or by coincidentally designing for the same use case. As a result, a portion of the examples in SketchGraphs may be considered to be duplicates, depending on the precise notion of duplicate adopted. For example, two sketches may exhibit visually similar geometry yet differ in their ordering of construction operations, placement in the coordinate plane, or constraints.

First, we normalize each sketch via centering and uniform rescaling. The numerical parameters in the primitive sequence are quantized to six bits, mapping their coordinates to a $64 \times 64$ grid. Then, for any set of sketches with identical sequences of primitives (including their quantized parameters), we keep only one, resulting in a collection of 1.7 million unique sketches. This procedure removes all sketches that have approximately equivalent geometry with the same ordering of construction operations.

### 4.2 BASELINES

We evaluate several ablation and conditional variants of our model as well as four baselines: 1) A uniform sampler that selects tokens with equal probability at each time step. 2) An LZMA compres-

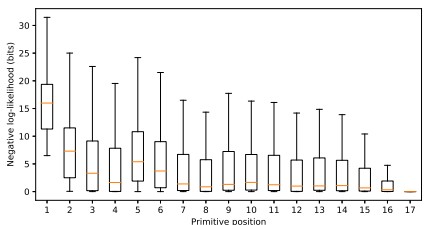

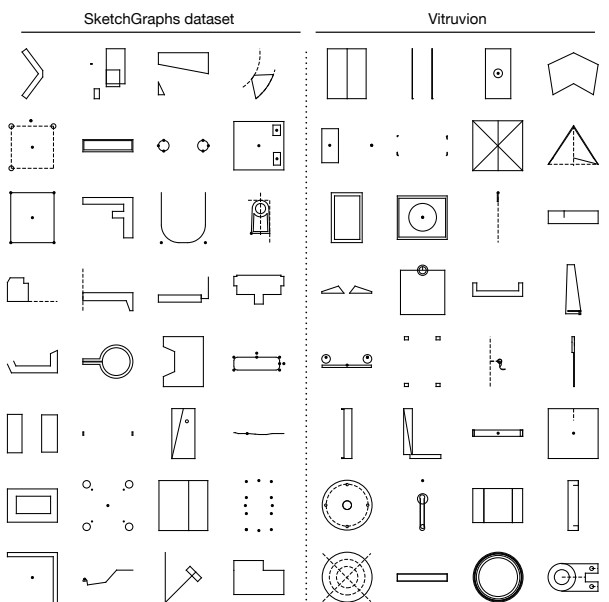

Figure 5: Average NLL per primitive position for the unconditional primitive model. In general, the model achieves lower NLL for primitives later in a sequence, as ambiguity decreases. However, a staircase pattern of length four is also apparent, due to the high prevalence of rectangles.

|  | Bits/Prim | Bits/Sketch | Accuracy |
|---|---|---|---|
| Uniform | 33.41 | 350.2 | 1.3 % |
| Compression | 15.34 | 160.7 | — % |
| Unconditional | 6.35 (0.01) | 66.6 (0.1) | 71.7 (0.0)% |
| Unconditional (perm.) | 8.15 (0.39) | 85.5 (4.1) | 60.3 (1.8)% |
| Image-conditional | 3.50 (0.05) | 35.1 (0.5) | 82.2 (0.2)% |

Table 2: Primitive model evaluation. Negative test log-likelihood in bits/primitive and bits/sketch. Next-step prediction accuracy (per-token avg.) is also provided. Standard deviation across 5 replicates in parentheses.

Figure 4: Random examples from the SketchGraphs dataset (left) and random samples from our unconditional primitive model (right).

sion baseline (see Section F). 3) The models proposed in Seff et al. (2020) (see Section G). 4) The models proposed in Willis et al. (2021) (see Section H).

## 4.3 PRIMITIVE MODEL PERFORMANCE

**Unconditional model.** The unconditional variant of our model is trained to approximate the distribution over primitive sequences without any context. As shown in Table 2, the unconditional primitive model achieves a substantially lower NLL than both the uniform and compression baselines. We employ nucleus sampling (Holtzman et al., 2020) with cumulative probability parameter of $p = 0.9$ to remove low-probability tokens at each step. Fig. 4 displays a set of random samples.

To assess how the predictive performance of the model depends on the sequence order, we train an unconditional primitive model on sequences where the primitives are shuffled, removing the model's exposure to the true ordering. As shown in Table 2, this results in a decline on next-step prediction. Importantly, the erasure of primitive ordering removes useful local hierarchical structure (e.g., high-level constructs like rectangles; see Fig. 5) as well as global temporal structure.

**Image-conditional generation.** Table 2 displays the performance of the basic image-conditional variant of the primitive model. By taking a visual observation of the primitives as input, this model is able to place substantially higher probability on the target sequence.

In Table 3, we evaluate the performance of several image-conditional models on a set of human-drawn sketches. These sketches were hand-drawn on a tablet computer in a $128 \times 128$-pixel bounding box, where the drawer first eyeballs a test set sketch, and then attempts to reproduce it. Likewise, each image is paired up with its ground truth primitive sequence, enabling log-likelihood evaluation.

We test three versions of the image-conditional model on the hand drawings, each trained on a different type of rendering: precise renderings, renderings from the hand-drawn simulator, and renderings with random affine augmentations of the hand-drawn simulator. Both the hand-drawn simulation and augmentations substantially improve performance. Despite never observing a hand drawing during training, the model is able to effectively interpret these. Figure 6 displays random samples from the primitive model when conditioned on images of hand-drawn sketches.

**Primer-conditional generation.** We assess the model's ability to complete primitive sequences when only provided with an incomplete sketch. Here, we randomly select a subset of test set sketches, deleting 40% of the primitives from the suffix of the sequence. The remaining sequence prefix then serves to prime the primitive model.

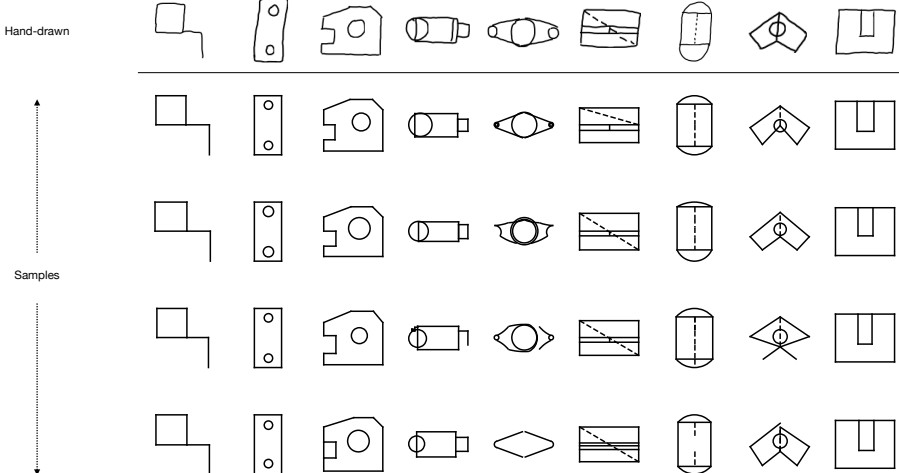

Figure 6: Image-conditional samples from our primitive model. Raster images of real hand-drawn sketches (top) are input to the primitive model. Four independent samples are then generated for each input. The model makes occasional mistakes, but tends to largely recover the depicted sketch with only a few samples.

| Training regimen | Bits/Prim | Bits/Sketch | Accuracy |
|---|---|---|---|
| Precise rendering | 22.80 | 292.3 | 47.0% |
| Hand-drawn augmentation | 10.59 | 135.5 | 65.5% |
| Hand-drawn + affine | 6.14 | 78.6 | 75.9% |

Table 3: Image-conditional primitive models evaluated on real hand-drawn sketches. NLL and predictive accuracy (per-token avg.) metrics are shown for three models trained with different data augmentation schemes.

| Model | Noiseless Testing | | Noisy Testing | |
|---|---|---|---|---|
| | Perplexity | Accuracy | Perplexity | Accuracy |
| Uniform | 4.984 | 0.6% | 4.984 | 0.6% |
| SketchGraphs | 0.246 | –% | 0.900 | –% |
| Noiseless training | 0.184 | 91.5% | 0.903 | 68.6% |
| Noisy training | 0.198 | 90.3% | 0.203 | 90.6% |

Table 4: Constraint model evaluation with per-token perplexity and accuracy. A model trained on noiseless data degrades substantially in the presence of noise.

Fig. 7 displays random examples of priming the primitive model with incomplete sketches. Because just over half of the original primitives are in the primer, there is a wide array of plausible completions. In some cases, despite only six completions for each input, the original sketch is recovered. We envision this type of conditioning as part of an interactive application where the user may query for $k$ completions from a CAD program and then select one if it resembles their desired sketch.

### 4.4 CONSTRAINT MODEL PERFORMANCE

We train and test our constraint model with two different types of input primitive sequences: noiseless primitives and noise-injected primitives. Noise-injection proves to be a crucial augmentation in order for the constraint model to generalize to imprecise primitive placements (such as in the image-conditional setting). As shown in Table 4 and Fig. 8, while both versions of the model perform similarly on a noiseless test set, the model trained according to the original primitive locations fails to generalize outside this distribution. In contrast, exposing the model to primitive noise during training substantially improves performance.

**Sketch editing.** Figure 2 illustrates an end-to-end workflow enabled by our model. Our model is first used to infer primitives and constraints from a hand-drawn sketch. We then show how the resulting solved sketch remains in a coherent state due to edit propagation.

## 5 CONCLUSION AND FUTURE WORK

In this work, we have introduced a method for modeling and synthesis of parametric CAD sketches, the fundamental building blocks of modern mechanical design. Adapting recent progress in modeling irregularly structured data, our model leverages the flexibility of self-attention to propagate sketch information during autoregressive generation. We also demonstrate conditional synthesis capabilities including converting hand-drawings to parametric designs.

Avenues for future work include extending the generative modeling framework to capture the more general class of 3D inputs; including modeling 3D construction operations and synthesis conditioned on noisy 3D scans or voxel representations.

## 6 ETHICS STATEMENT

The goal of this work is to automate a tedious process and enable greater creativity in design, but we acknowledge the potential societal impact that could arise from any elimination of human labor. Moreover, there is the potential risk in high-stakes applications for automatically synthesizing mechanical systems that do not meet human structural standards. We view this work, however, as being part of a larger human-in-the-loop process which will still involve human expertise for validation and deployment.

## 7 REPRODUCIBILITY STATEMENT

In addition to the model descriptions in the main text, we have provided all details of the training procedure (optimizer, learning rate, hyperparameters etc.) in Appendix E. For code and pre-trained models, see `https://lips.cs.princeton.edu/vitruvion`.

### ACKNOWLEDGMENTS

The authors would like to thank Yaniv Ovadia and Jeffrey Cheng for early discussions of this project. Thanks to all members of the Princeton Laboratory for Intelligent Probabilistic Systems for providing valuable feedback. Additionally, we would like to thank Onshape for the API access as well as the many Onshape users who created the CAD sketches comprising the SketchGraphs training data. This work was partially funded by the DataX Program at Princeton University through support from the Schmidt Futures Foundation and by the NSF Institute for Data-Driven Dynamical Design (NSF 2118201).

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

# Appendices

## A ADDITIONAL EVALUATION

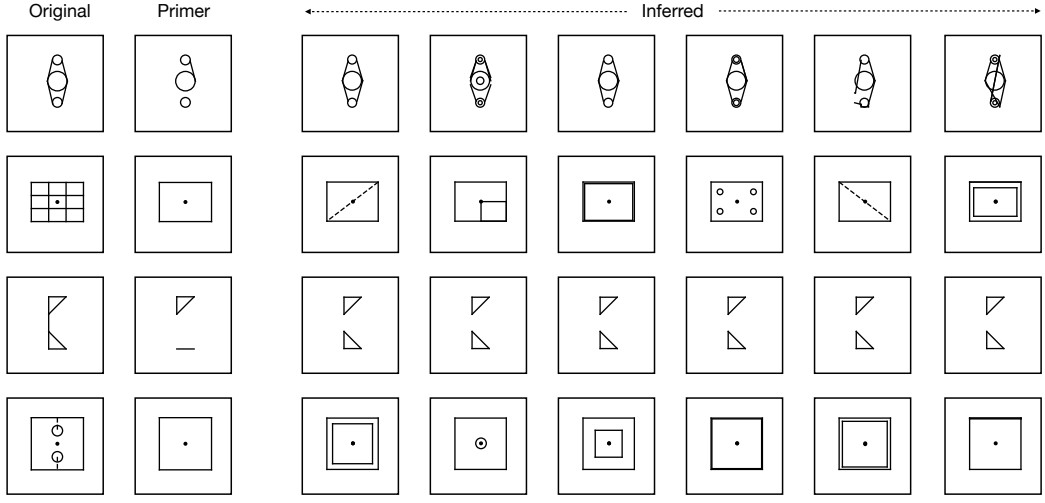

Figure 7: Primer-conditional generation. We take held-out sketches (left column) and delete 40% of their primitives, with the remaining prefix of primitives (second column from left) serving as a primer for the primitive model. To the right are the inferred completions, where the model is queried for additional primitives until selecting the stop token. Note that only the primitive model is used in these examples.

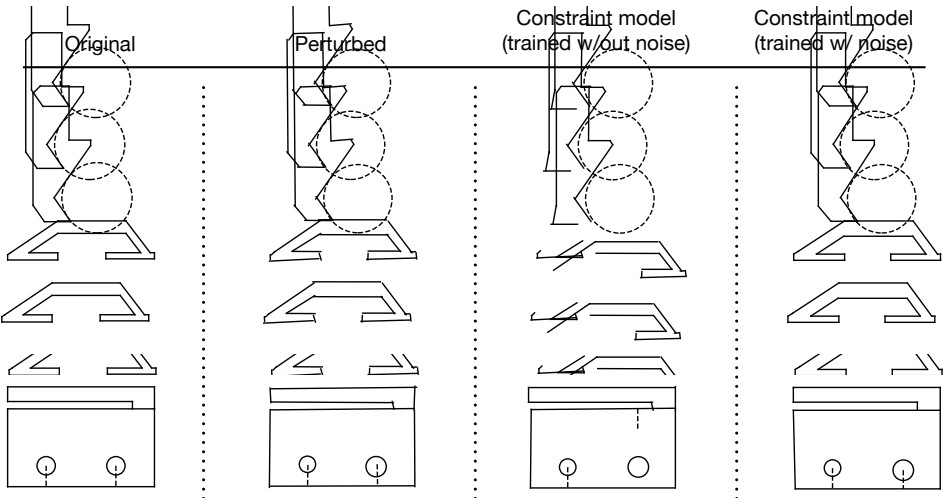

Figure 8: Constraining primitives. We take test set sketches (leftmost column) and perturb the coordinates of the primitives with Gaussian noise (second from left). The constraint model trained without noise (second from right) often fails to correct the primitive locations, in constrast to the constraint model trained on noise-augmented data (rightmost column).

### A.1 DISTRIBUTIONAL STATISTICS

We quantitatively compare distributional statistics for generated sketches and those from the test set. 10K sketches are sampled from our full unconditional model (both primitives and constraints). In particular, we compare the number of generated primitives, constraints, total degrees of freedom, degrees of freedom removed by constraints and net degrees of freedom across generated sketches in Fig. 9.

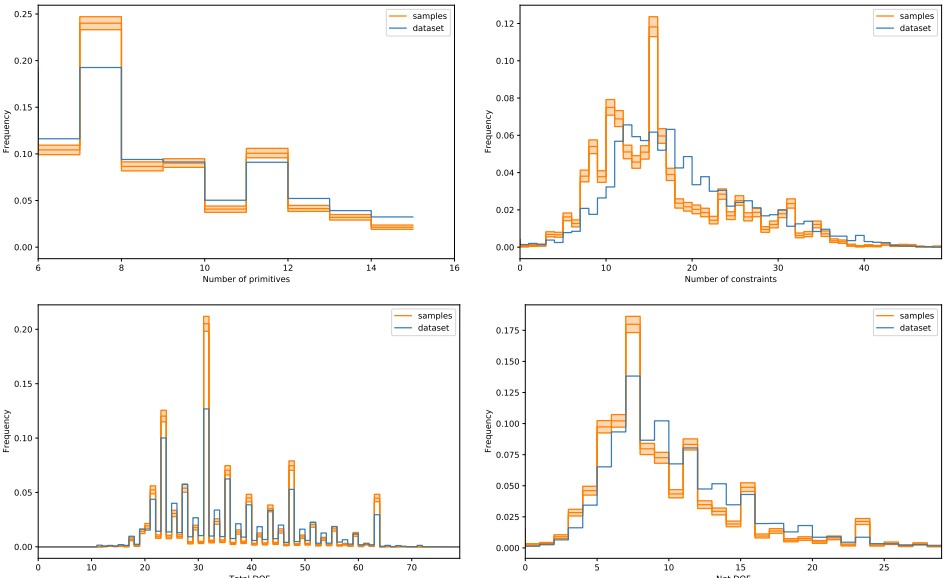

Figure 9: Comparison of various statistics between between generated and dataset sketches. Shaded regions represent bootstrapped 95% confidence intervals.

## A.2 ABLATION: TEST-TIME ORDERING SHUFFLING

|  | Bits/Prim | Bits/Sketch | Accuracy |
|---|---|---|---|
| Uniform | 33.41 | 350.2 | 1.3 % |
| Compression | 15.34 | 160.7 | — % |
| Unconditional | 6.35 (0.01) | 66.6 (0.1) | 71.7 (0.0)% |
| Unconditional (perm.) | 8.15 (0.39) | 85.5 (4.1) | 60.3 (1.8)% |
| Unconditional (perm. test) | 9.98 (0.03) | 104.7 (0.3) | 55.5 (0.2)% |
| Image-conditional | 3.50 (0.05) | 35.1 (0.5) | 82.2 (0.2)% |

Table 5: Additional evaluations of the primitive model. Unconditional (perm. test) denotes the primitive model trained on standard data but evaluated on a randomly permuted dataset.

We conduct an additional ablation in order to better complement our understanding of the dependency of the primitive model on the sequence order in Table 5. We consider the standard unconditional model, but evaluated on randomly permuted primitive sequences. We observe that this leads to a performance which is significantly worse (NLL of 104.7 bits/sketch compared to 66.6 bits/sketch on the original test set), underlining the fact that our model learns significant information about the primitive sequence order.

## A.3 TYPICAL FAILURE CASES

For the hand-drawn conditional primitive model, a typical failure case involves non-axis aligned line segments (e.g., columns 5 and 8 of Fig. 6), which are less prevalent in the data than axis-aligned ones. Data augmentation tailored to this scenario may be one route to alleviate this. For the constraint model, one typical failure case is when the model selects an incorrect reference, particularly when deployed on noise-injected primitives. We found using a tighter nucleus sampling parameter (0.7) to mitigate this substantially. We imagine our model serving as part of an interactive application where a user can select from a handful of output sketches, mitigating the effect of an occasional incorrect prediction.

## B  HAND-DRAWN NOISE MODEL

The noise model renders lines as drawn from a zero-mean Gaussian process with a Matérn-3/2 kernel that have been rotated into the correct orientation. Arcs are rendered as Matérn Gaussian process paths in polar coordinates with angle as the input and the random function modulating the radius. No additional observation noise is introduced beyond the standard "jitter" term. The length-scale and amplitude are chosen, and the Gaussian process truncated, so that scales are consistent regardless of the length of the line or arc.

## C  SYSTEM DETAILS

| Value Token | Primitive Type |
|---|---|
| 0 | Pad |
| 1 | Start |
| 2 | Stop |
| 3 | Arc |
| 4 | Circle |
| 5 | Line |
| 6 | Point |

| Value Token | Constraint Type |
|---|---|
| 0 | Pad |
| 1 | Start |
| 2 | Stop |
| 3 | Coincident |
| 4 | Concentric |
| 5 | Equal |
| 6 | Fix |
| 7 | Horizontal |
| 8 | Midpoint |
| 9 | Normal |
| 10 | Offset |
| 11 | Parallel |
| 12 | Perpendicular |
| 13 | Quadrant |
| 14 | Tangent |
| 15 | Vertical |

Table 6: Mapping for both primitive types (left) and constraint types (right) to the reserved set of value tokens for each model. The tokens 0, 1, and 2 are used in both models for padding, start, and stop, respectively.

Primitive and constraint types are represented using a pre-specified range of each model's value tokens. For example, for primitives, as described in the Tokenization paragraph of Section 3.1, a small set of the value tokens (four tokens for Arc, Line, Circle, and Point) are reserved for type specification. Three value tokens are also reserved for Start, Stop, and Pad.

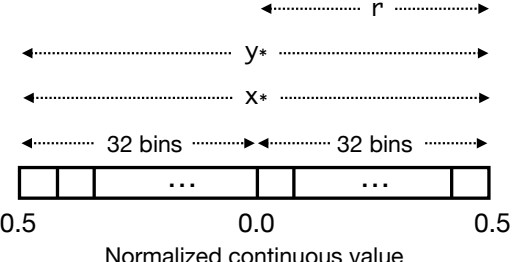

Figure 10: Utilization of quantization bins by the different numerical parameters. $x$ and $y$ coordinates may occupy the full range of bins (representing both negative and positive values) but the radius parameter only utilizes half of the bins (32 in our case) since it must be positive. This means the numerical interval represented by each bin is invariant to which parameter is being considered.

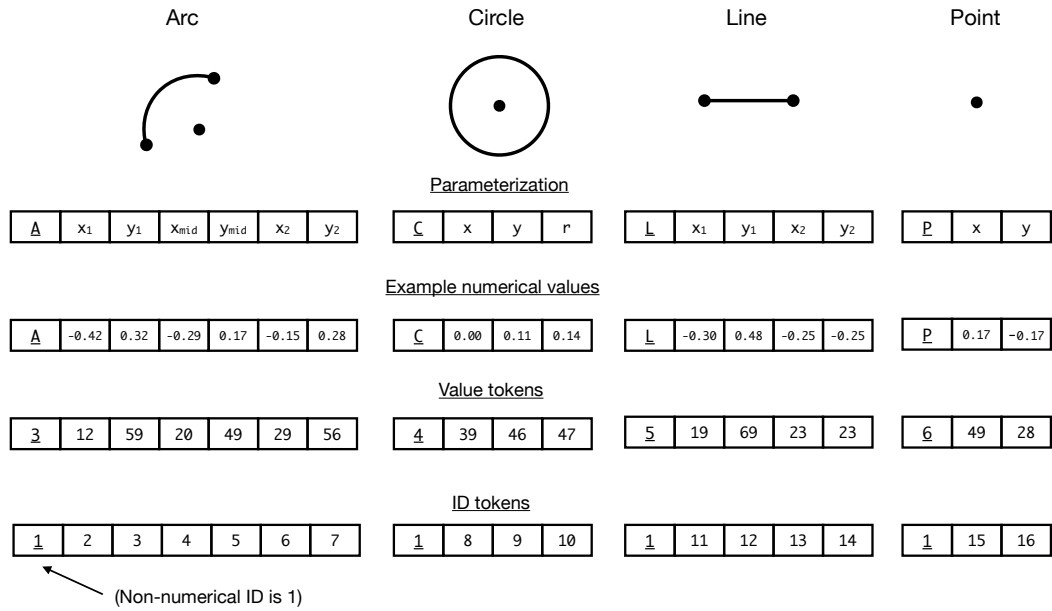

Figure 11: Mapping of primitive parameters to value and ID tokens. Example raw numerical values and the resulting value token that gets assigned to it are shown for each primitive parameter.

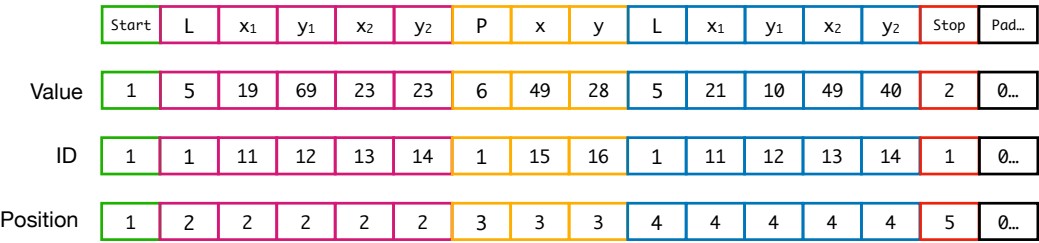

Figure 12: An example primitive sequence (consisting of a line, a point, and another line) and its associated sequences of value, ID, and position tokens.

**Constraint model output.** The constraint model's output consists of both categorical types and references to primitives. At any given step during inference, the output dot product is computed between the decoder's output representation and the concatenation of the learned embeddings for the constraint types and the (sub-)primitive encodings. That is, the softmax output layer of the constraint model selects from the union of both categoricals (for the constraint types) and the references. We may view this still as producing a pointer, but the possible targets for the pointer are not only the primitive references, but the set of constraint types as well.

**Primitive and sub-primitive referencing.** The constraint model must be able to reference both individual primitives or subcomponents thereof (sub-primitives, consisting of specific points that may be the targets of constraints). We took a simple approach where, for each primitive type, a subset of the tuple of learned encodings (as output by the primitive model) are tasked with representing either the entire primitive or a specific sub-primitive. For example, for a line, which consists of a tuple of length 5 (type Line, x0, y0, x1, y1), we use the encoding at index 0 to represent the line segment as a whole, index 1 (the x coordinate of the first endpoint) to represent the first endpoint, and index 3 (the x coordinate of the second endpoint) to represent the second endpoint. We note that, after several layers of transformations in the primitive model, the representations at these indices do not solely represent the original token but instead learn to incorporate the information needed to represent the (sub-) primitive they were assigned. Note that this mapping is arbitrary and there are other design routes that could produce equivalent results.

For completeness, we list the reference indices for each primitive type below where the index indicates which element of the primitive's tokenization is given a specific representation responsibility:

- Arc: index 0 (whole arc), index 1 (first endpoint), index 3 (centerpoint), index 5 (second endpoint)
- Circe: index 0 (whole circle), index 1 (centerpoint)
- Line: index 0 (whole line segment), index 1 (first endpoint), index 3 (second endpoint)
- Point: index 0 (whole point)

## D    QUANTIZATION STUDY

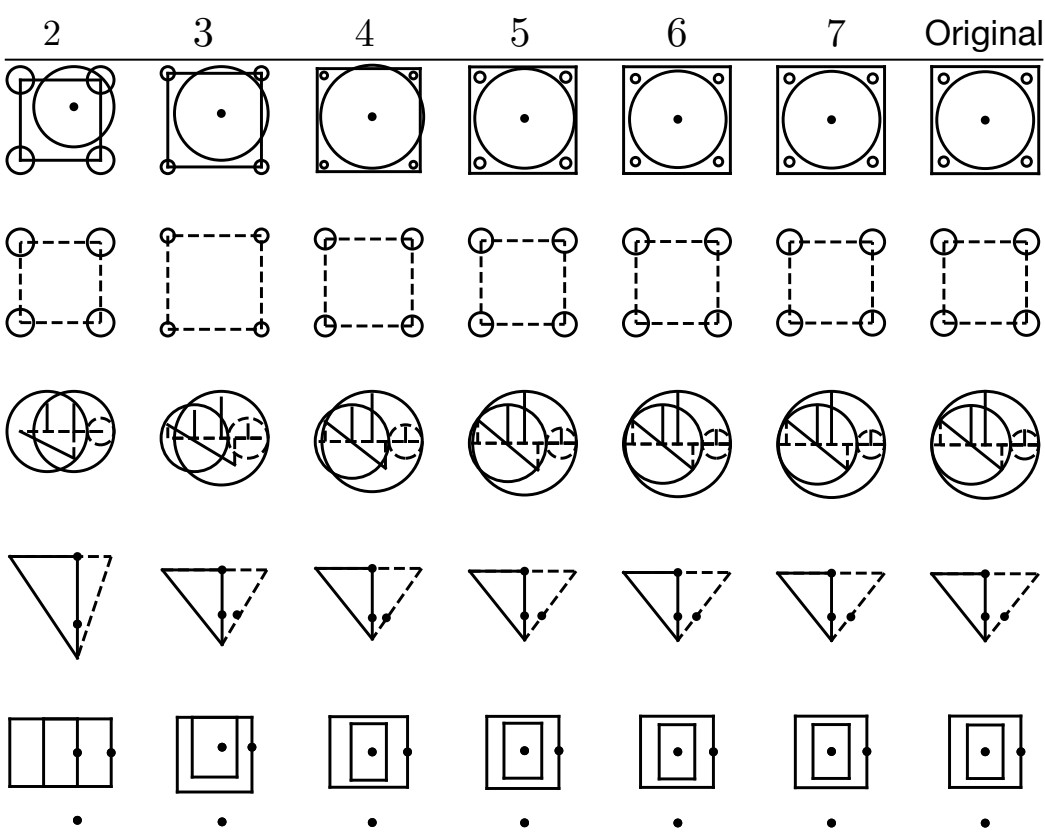

Figure 13: Reconstruction of SketchGraphs sketches at various quantization bit widths. Bit widths from 2 bits (4 unique parameter values, leftmost column) to the original single-precision floating-point representation (rightmost column) are illustrated above. The quantization used in our model is 6 bits.

## E    EXPERIMENTAL DETAILS

This section provides further details of the model architecture and training regime.

### E.1    MODEL ARCHITECTURE

Our models all share a main transformer responsible for processing the sequence of primitives or constraints which is the target of the inference problem. This transformer architecture is identical across all of the models, using a standard transformer decoder with 12 layers, 8 attention heads, and a total embedding dimension of 256. No significant hyper-parameter optimization was performed. The raw primitive generative model is only equipped with the decoder on the primitive sequence, and is trained in an autoregressive fashion. The constraint model, in addition to the transformer decoder

on the constraint sequence, also performs standard cross-attention into an encoded representation of the primitive sequence.

The image-to-primitive model is additionally equipped with an image encoder based on a vision transformer (Dosovitskiy et al., 2021), which is tasked with producing a sequence of image representations into which the primitive decoder performs cross-attention. We use $128 \times 128$-sized images as input to the model. Non-overlapping square patches of size $16 \times 16$ are extracted from an input image and flattened, producing a sequence of 64 flattened patches. Each then undergoes a linear transformation to the model's embedding dimension (in this case, 256) before entering a standard transformer encoder. This transformer encoder has the same size as the decoder used in our other models.

## E.2 TRAINING

Training was performed on cluster servers equipped with four Nvidia V100 GPUs. Total training time was just under 2 hours for the primitive model, and 5.5 hours for the image-to-primitive model and the constraint model. The primitive model and the constraint model were each trained for 30 epochs, and the image-to-primitive model was trained for 40 epochs, although the total number of epochs did not appear to cause significant differences in performance. The models were trained using the Adam optimizer with "decoupled weight decay regularization" (Loshchilov & Hutter, 2019), with the learning rate set according to a "one-cycle" learning rate schedule (Smith & Topin, 2018). The initial initial learning rate was set to 3e-5 (at reference batch size 128, scaled linearly with the total batch size). The batch size was set according to the memory usage of the different models at 1024 / GPU for the raw primitive model, 512 / GPU for the image-to-primitive model, and 384 / GPU for the constraint model.

## E.3 IMAGE-TO-PRIMITIVE DATA

Prior to training the image-to-primitive model, we generate samples from our hand-drawn noise model (see Section B) in a separate process. Five samples are generated for each sketch in the dataset, taking a total of approximately 160 CPU-hours. This process is parallelized on a computing cluster. During each training epoch, one random image (out of the five generated samples) is selected to be used by the model. An additional data augmentation process is used during training, whereby random affine transformations are applied to the images. The augmentation is chosen randomly among all affine transformations which translate the image by at most 8 pixels, rotate by at most 10 degrees, shear the image by at most 10 degrees, and scale the image by at most 20%.

## F COMPRESSION BASELINE

The compression baseline is computed by first tokenizing primitives according to the tokenization scheme described in section 3.1, representing the token sequences as 6-bit integers, and compressing the given data using the LZMA compression algorithm. The compression is performed over the entire dataset (including training, validation and test splits), and the reported quantities are averages over the entire dataset. In particular, note that we may expect the entropy rate of the data to be lower than reported.

## G SKETCHGRAPHS BASELINE

We adapt the models proposed in Seff et al. (2020) as a baseline comparison for some of the tasks studied in this paper. Namely, we evaluate the proposed autoconstrain model for the constraint inference task (conditional on primitives), and the proposed generative model on joint unconditional modelling of the primitives and constraints. We do not attempt to adapt these baseline models to the image-conditional or primer-conditional generation tasks also studied in the current work. In order to ensure a fair comparison, we adapt the SketchGraphs models to operate on the featurization used in the current work, rather than the one proposed in Seff et al. (2020), as described in Sections 3.1 and 3.2.

### G.1 GENERATIVE MODEL

We consider an autoregressive graph neural network as a raw generative model on the joint sequence of primitives and constraints as described in (Seff et al., 2020, Section 4.2). We note that by its nature, the graph neural network operates jointly on the primitives (nodes) and constraints (edges),

and thus corresponds to a different factorization of the distribution of the sketch as the one proposed in the current work.

As the SketchGraphs model does not incorporate continuous primitive parameters such as position, we make some adaptations in order to model the same data as in the current work. The continuous parameters for primitives are input according to the procedure described in (Seff et al., 2020, Appendix D.2), using the quantization scheme in the current work. They are generated according to the scheme described for constraints in (Seff et al., 2020, Appendix E.1). In addition, we filter out constraints with continuous parameters as these are not modelled by the current work.

We obtain an average log-likelihood of 124.0 bits per sketch for the SketchGraphs generative model. By combining the primitive model and the constraint model presented in this paper (Vitruvion), we obtain instead an average log-likelihood of 85.3 bits per sketch (66.6 bits per sketch for the primitives, and 18.7 bits per sketch for the constraints conditioned on the primitives).

### G.2 CONSTRAINT MODEL

We adapt the "autoconstrain" model proposed in Seff et al. (2020) and compare it to our primitive-conditional constraint model. We also note here that the model operates on a somewhat different factorization of the sequence of constraints, and thus we may only compare log-likelihood at the sketch level (i.e., for inferring all constraints in a sketch, see Table 4). For presentation purposes, we have rescaled that value to an equivalent per-token value, although the SketchGraphs model does not use the same tokenization, and hence no comparable accuracy metric can be computed.

## H  QUALITATIVE BASELINES

**Seff et al. (2020).** Samples from the SketchGraphs generative model (Seff et al., 2020) may be viewed in Figure 11 from their work. Because this model does not learn to output primitive coordinates, it solely relies on the constraint graph and Onshape solver to determine the final configuration. However, solving geometric constraints under poor coordinate initialization is difficult, and likewise the solver often fails for their model, limiting the sophistication of sketches that are generated.

Figure 5 from Seff et al. (2020) demonstrates autoconstraining and then editing a sketch. This is very similar to our own Fig. 2. It is difficult to draw any conclusions about the qualitative performance as only one example is shown in their paper. They do include a few dimensional constraints (e.g., distance constraint) in their model's targets, while in our work we model all categorical constraints.

**Willis et al. (2021).** Two models are introduced in Willis et al. (2021), CurveGen and TurtleGen. Both solely model primitives without outputting any constraints. CurveGen, has a two-stage factorization, like unconditional Vitruvion, except both stages are used for building primitives. It first outputs vertex locations, and then a second model groups these vertices into primitives (line etc.). TurtleGen instead produces sketches as a single sequence of pen drawing commands (within a graphics program) in order to form closed loops. Both models utilize architectures based on transformers, similar to ours.

Figures 5, 8, and 9 from Willis et al. (2021) display some samples from their two models. Subjectively, the TurtleGen samples tend to appear similar to the SketchGraphs model samples mentioned above, but with a bit more variety. CurveGen samples appear much more sophisticated, exhibiting the kinds of symmetries found in the SketchGraphs dataset. However, there are some sketches that exhibit a lack of coherence, where points that would otherwise be coincident are not (e.g., Fig. 5, second to last row, Fig. 9 second row). We note that Willis et al. (2021) does not model construction primitives while we do (they appear as dashed lines in our renderings), so a direct visual comparison is slightly muddled by that.

Willis et al. (2021) showcases how the auto-constrain tool of AutoCAD can apply parallel and perpendicular constraints to their output sketches in Figure 7. While this is a useful tool, it requires manually setting the priority and tolerance for each individual constraint type. Instead, in our work, we propose a learned model that can adapt to the context of a given design automatically without the user toggling these settings.

None of the models above target image-conditional generation, which we explore in our work (including conditioning on hand-drawings).

