# OpenReview forum: "Vitruvion: A Generative Model of Parametric CAD Sketches"
_ICLR.cc/2022/Conference — ICLR 2022 Poster_

### Official Review · Reviewer_9htv · 2021-10-21

**Correctness:** 4
**Technical Novelty And Significance:** 4
**Empirical Novelty And Significance:** Not applicable
**Recommendation:** 8
**Confidence:** 4

**Main Review:**

**Strengths**
- The paper is well written, easy to follow. Section 2 provides a clear positioning of the work against prior efforts on similar problems.
- The method itself is well motivated, the proposed architecture makes sense and is described in sufficient details.
- The qualitative and quantitative evaluations suggest that the model performs well on the task.

**Weaknesses**
- I would have appreciated additional visual results on the task of inferring constraints over existing sketches (Fig.8) and on the task of generating CAD sketches from hand-drawn sketches (Fig.6). While the few results shown are very encouraging, I would have liked a discussion of typical failure cases (since I suspect some failure on such ambiguous tasks).
- My main concern about this paper is that it appears very similar to [Ganin et al. 2021, Para et al. 2021], which will both appear at Neurips. Checking the ICLR reviewer instructions, I see that these papers are indeed considered concurrent (*if a paper was published (i.e., at a peer-reviewed venue) on or after June 5, 2021, authors are not required to compare their own work to that paper.*), which I interpret as not being a reason for rejecting this submission.
Still, from a reader perspective, I would have liked a detailed discussion of [Ganin et al. 2021, Para et al. 2021] to understand if they are indeed very similar (both target the same problem, both rely on a Transformer+PointerNetwork architecture), or if the proposed model introduces significant differences. The submission mentions that these two concurrent papers do not attempt to train/test on hand-drawn / noisy input, but this seems more of a detail in the way the method is used than a fundamental difference in the actual model.


**Summary Of The Paper:**

This paper presents a model capable of generating CAD sketches with constraints. Such sketches are typically formed as a sequence of 2D primitives (points, lines, arcs, circles) related by geometric constraints (parallelism, orthogonality, coincidence, etc). The model generates these sequences in an auto-regressive fashion, one primitive at a time, and then one constraint at a time. The model is trained on a large dataset of CAD sketches [Seff et al. 2020] that contains the necessary ordered sequences of CAD primitives and constraints.

The model is implemented as two Transformers, one to generate the sequence of primitive, and one to generate the sequence of constraints conditioned on the primitives. For the second part, a PointerNetwork mechanism is used to identify the primitives referenced by each constraint.

The model is used either to generate random CAD sketches that exhibit similar statistics as the training data (as measured by the number of primitives and the number of constraints used), or to auto-complete a partial CAD sketch by predicting missing primitives and constraints. In addition, an image-conditioned version of the model is also described, which can take as input an approximate bitmap drawing and turn it into a constrained CAD sketch.

**Summary Of The Review:**

If the similarity to the concurrent work of Ganin et al. 2021 and Para et al. 2021 should not be accounted for, then I am in favor of acceptance because the described model is sound and effective. But I remain uncomfortable accepting a paper that shares so many similarities with papers that have already been reviewed and accepted, even if shortly before the deadline. I suspect that the authors have been inspired by these concurrent works, maybe in the development of their method, its description, its evaluation, which is why I have a hard time ignoring them in my evaluation.

To be clear, if [Ganin et al. 2021] and [Para et al. 2021] are to be ignored, then the technical novelty is of level 4, but if they should be considered, then the technical novelty drops to 1 or 2 and I would argue for rejection.

--- Additional questions ---

Since the constraint model is trained separately from the primitive model, is it trained with ground-truth primitive sequences from Seff et al., or from the (approximate) sequences predicted by the primitive model? Section 3.2 mentions that the constraint model must account for potentially imperfect generation from the primitive model, but I didn’t fully understand how this is achieved. Is it only achieved by perturbing the primitive parameters with Gaussian noise? What if the predicted primitive type is wrong, wouldn’t such error degrade all subsequent predictions?

I appreciate the practical details provided in Section 3.1 and 3.2, and in Appendix C. Yet, I would have liked a diagram of the two Transformers to better understand the flow of information, and to illustrate how the sequences are fed to the networks. This would be particularly useful for the constraint prediction network, as the mechanism to cross-attend the primitive sequence and to point to its elements remains a bit vague to me.

I didn’t fully understand the tokenization in Section 3.1. Since the ID token specifes the parameter type, why not also use it to specify the primitive type rather than allocate part of the value token for this purpose?

---

> ### Public Comment · ~Yaroslav_Ganin1 · 2021-11-09
> **A remark**
>
> I'm one of the authors of [Ganin et al. 2021]. I don't have a clear understanding of the ICLR's prior work policy but I would like to point out that our work first appeared on arXiv on the 6th of May and to my knowledge this is the only publicly available version of the paper at the moment. Presumably, the authors of this submission refer to this revision rather than to the NeurIPS one.

---

> ### Author Response · Authors · 2021-11-23
> **Response to Reviewer 9htv**
>
> Thanks, Reviewer 9htv, for your thoughtful comments. We address them below.
>
> *"While the few results shown are very encouraging, I would have liked a discussion of typical failure cases (since I suspect some failure on such ambiguous tasks)."*
>
> We have added some discussion of typical failure cases for both the hand-drawn conditional model and the constraint model. For the hand-drawn conditional model, one typical failure case involves non-axis aligned line segments (e.g., Figure 6, columns 5 and 8), which are less prevalent in the data than axis-aligned ones. For the constraint model, one failure case is when the model selects an incorrect reference. We found this to be substantially mitigated when using a smaller nucleus sampling parameter (0.7).
>
>
> *"Since the constraint model is trained separately from the primitive model, is it trained with ground-truth primitive sequences from Seff et al., or from the (approximate) sequences predicted by the primitive model?"*
>
> It is trained by conditioning on the ground truth primitive sequences from Seff et al., 2020, analogous to teacher forcing, but with Gaussian noise added to the coordinates.
>
>
> *"What if the predicted primitive type is wrong, wouldn’t such error degrade all subsequent predictions?"*
>
> If the predicted primitive type is wrong at a given step (e.g., in an image-conditional setting), this can certainly degrade the remaining predictions of the sequence. This is also true for a standard language model; an incorrect word prediction can degrade the rest of an output sentence. We imagine our model serving as part of an interactive application where a user can select from a handful of output sketches, mitigating the effect of an occasional incorrect prediction.
>
>
> *"Since the ID token specifes the parameter type, why not also use it to specify the primitive type rather than allocate part of the value token for this purpose?"*
>
> There are several reasonable routes one could take for tokenization. In our case, we found it simplest to have the value tokens serve as a complete description of the primitive/constraint sequences. Then, we augment this input with additional annotations, namely ID tokens and position tokens, solely for the transformers' benefit as they would otherwise be blind to the ordering. Similar to Nash et al., we apply position tokens at the primitive/constraint level (i.e., all tokens for a given primitive share have equivalent position tokens) and then apply ID tokens to further disambiguate each primitive's/constraint's parameters.

---

> > ### Comment · Reviewer_9htv · 2021-11-28
> > **In favor of acceptance**
> >
> > I remain unconfortable seeing similar ideas written in 3 or 4 different papers, as it ultimately results in duplicate work for authors, reviewers and readers. But I understand that this is the state of a field that is expanding greatly and is moving very quickly.
> >
> > Apart from that issue, which is not proper to this submission, I am positive about the paper, and I appreciate the response to the concerns raised by the other reviewers. I do think that this familly of papers represent an exciting step towards automated CAD modeling, and I hope it will stimulate further progress to reach practical applications. I thus maintain my original recommendation (8: accept, good paper)

---

### Official Review · Reviewer_rqPX · 2021-11-01

**Correctness:** 4
**Technical Novelty And Significance:** 3
**Empirical Novelty And Significance:** 3
**Recommendation:** 8
**Confidence:** 4

**Main Review:**

It is somehow natural that after a release of an exciting new dataset multiple groups are trying to target core questions that can be tackled due to its release. There is already important work in this area, i.e. Ganin et al. (2021) and Para et al. (2021), that has substantial overlap with this submission. According to the rules of ICLR, I believe this previous work should count as independent publication. The authors also position their work as concurrently developed.

Overall, a suitable engineering choice is to model sketches as sequences of tokens and to employ multiple transformers for sequence modeling. A strong inspiration for this (as well as the other papers) was PolyGen from Nash 2020 which is appropriately cited multiple times in the paper. Working with transformers to model sequences is difficult work and it typically takes quite a bit of skill and work to achieve good results, such as the ones shown in the paper. Working with such large datasets is also difficult in general.

The paper is generally very well written (language wise) and nicely illustrated. My main request for improvement is to add details in the writing. I would be interested to know more details of
*) the architecture and training
*) the data formats for sequence modeling (input as well as output)
There is a promise to release code, which should help clarify some details in the future. However, I would strongly request that these details are included in the paper. For example, the paper proposes to use references to primitives and sub-primitives. It is unclear how exactly this is done. The tokenization in 3.1 and 3.2 could be explained better with examples and the precise number of bits and precise number of choices that are being encoded. It seems that Ganin et al. and Para et al. have more content devloted to clarify details, but this submission does not have a corresponding appendix with details (e.g. appendix A and B in Ganin et al.). For example xCenter is an ID token, but what are all the other ones? What are the lookup table options for ID, Value, and Position (mainly ID and value are unclear). How many lookup tables (quantization tables) are there? Are they context dependent? There are multiple choices how exactly to do the encoding.

6-bit quantization appears a bit low and I wonder if that is really a suitable choice to achieve very good results. Maybe there could be an ablation study. Intuitively, I really doubt that 6-bit is enough to encode details. I do not see how the constraints can fix that large loss of information.

The paper does provide a list of reasonable results, but at the same time there could be more comparisons and ablation studies to put the work into context. The results feel maybe a bit less extensive than the other two papers. It would also be informative if the paper could add more comparison to the previous papers to be helpful to future researchers in the field (even though they should not be required to do so). Running other code is probably tricky, but writing a table or a list comparing some of the design decisions in the text should be feasible.

The work does a good job conditioning on images, a result that is not present in Ganin et al. and Para et al. and a unique feature of this work.

**Summary Of The Paper:**

The work builds on the SketchGraphs dataset that contains CAD sketches, consisting of primitives and constraints. This dataset opened many opportunities to tackle research problems that could not be easily be tackled before. One important question is how to define a generative model that can unconditionally sample from the distribution. Another important question is how to sample conditioned on some input, e.g. a hand drawn sketch. This paper looks at both of these questions.


**Summary Of The Review:**

Assuming the authors are willing to add (a lot) more details, I would suggest that this paper should be accepted.

---

> ### Author Response · Authors · 2021-11-23
> **Response to Reviewer rqPX**
>
> Thanks, Reviewer rqPX, for your thoughtful comments. We address them below.
>
> *"I would be interested to know more details of \*) the architecture and training \*) the data formats for sequence modeling (input as well as output)."*
>
> We have added further details and diagrams to the appendix to clarify these procedures.
>
>
> *"6-bit quantization appears a bit low and I wonder if that is really a suitable choice to achieve very good results."*
>
> We appreciate this question, as initially it was unclear to us what level of quantization would be required to reconstruct coherent sketches. While sketches can display sophisticated geometry, the regularity of the patterns as well as our normalization procedure (centering and rescaling) empirically lead to 6-bit quantization being sufficient. Note that all samples from the primitive model shown in the paper (e.g., Figure 4, right) use only 6-bit quantization. In addition, note that Ellis et al., 2018 (generating TikZ figures) constrains coordinates to a 16x16 grid and yet their model still produces fairly involved shapes. Our 6-bit quantization effectively snaps the sketches to a 64x64 grid, with 16 times the number of grid points compared to their quantization.
>
> In the interest of being thorough, we have included a a visual demonstration of quantizing SketchGraphs sketches at various bitwidths (Appendix D). The 6-bit quantization used in our work visually seems to be sufficient (subjectively) to capture much of the coordinate-level precision required for the data used. Finally, we emphasize that the quantization bit width can be treated as an implementation-specific parameter; it is conceivable that differing data sources may require different coordinate quantizations.

---

> > ### Comment · Reviewer_rqPX · 2021-11-27
> > **Remain with my assessment**
> >
> > I read the other reviews and the authors comments and remain with my assessment. My review did not explicitly discuss all the related and concurrent work. However, also considering other work brought up in other reviews, this work makes a clear case for a contribution over previous work. All work that overlaps the critical parts of the contribution is published concurrently.

---

### Official Review · Reviewer_DfnC · 2021-11-02

**Correctness:** 3
**Technical Novelty And Significance:** 2
**Empirical Novelty And Significance:** 3
**Recommendation:** 6
**Confidence:** 4

**Main Review:**

The task of using deep learning to generated structure representations of graphics useful in CAD applications is an interesting and open area of research that has potential for practical applications. The method proposed in this paper utilizes a reasonable approach based on state-of-the-art architectures and techniques. The paper is well-written, and the shown results look good given the dataset that is used for training.

My main concern about this paper has to do with its technical novelty. While it is true that the SketchGraphs dataset is fairly recent, and there have not been many papers published that utilize it, it is not completely obvious what are the high-level technical takeaways that can be useful for building on the proposed method. All of the components are certainly reasonable and demonstrate impressive engineering but seem like fairly standard applications of existing architectures. On the other hand, while the results are nice, I don't think they are ready to be put into a production-level system, in part due to the relative simplicity of the generated sketches. I would encourage the authors to clarify what is the broad novel insight for building learning-based systems that produce vectorized or CAD-style data and/or to provide some validation that the method can be practically helpful in a real-world CAD pipeline.

The other main issue that I see is with regards to comparison with related work. Some of the related work that's noted to be concurrent is in fact prior work. In particular, "Engineering Sketch Generation for Computer-Aided Design" [Willis et al. 2021] appeared at CVPR 2021 and, as the authors note, proposes a similar transformer-based CAD sketch generative model. It would be important to include a comparison to this work or at least a more detailed discussion. The authors mention that the main distinction is that Willis et al. do not produce the constraint graph as part of their output. This is true, but the evaluation of the learned constraint graphs is fairly limited---it is essentially only demonstrated in one small figure. Willis et al. show how the standard auto-constrain feature in AutoCAD can be used to infer constraints for sketches produced by their model. Do the learned constrain graphs here address certain cases that cannot just be handled by this sort of simple post-processing?

A more minor issue is that it would be very helpful to include a diagram showing the network architecture---while the text is clearly written, it is a bit hard to keep track of all the moving parts.

**Summary Of The Paper:**

This paper proposes a new deep generative model for 2D CAD sketches. The transformer-based model is able to generate both the primitives as well as the primitive constraints that make up a sketch, and its output can be then imported into standard CAD software. In addition to unconstrained sketch generation, the authors demonstrate experiments where they use the model to automatically complete partial sketches as well as produce sketches that resemble a rough human-drawn sketch that is given as input.

**Summary Of The Review:**

This paper is well-written and tackles an interesting problem in a sensible way. However, due to limited novel technical insight and comparison to past work, I think it requires some improvement before it is ready to be published.

---

> ### Author Response · Authors · 2021-11-23
> **Response to Reviewer DfnC**
>
> Thanks, Reviewer DfnC, for your thoughtful comments. We address them below.
>
> *"All of the components are certainly reasonable and demonstrate impressive engineering but seem like fairly standard applications of existing architectures... what are the high-level technical takeaways..."*
>
> While transformer-based architectures have become popular in the literature (especially for sequential domains) parametric CAD is very underexplored in the community and far from being a standard application. The constraint programs found in CAD exhibit significant differences from the original application domain for this class of models, namely natural language processing (NLP). Credibly accounting for these differences induces nontrivial adaptations on the original architectures, including the ability to produce explicit, rather than implicit, hierarchical references to achieve a multi-stage system such as the one we present.
>
> *"while the results are nice, I don't think they are ready to be put into a production-level system"*
>
> We do not intend to assert that our demonstrated system is production hardened, nor that it is sufficiently mature for full-scale industrial application. This is generally not the desired outcome for research published at a conference like ICLR. We demonstrate a route to building a learned CAD sketch generator (an important domain as you note), evaluating this generator on specific application scenarios unexplored in related work. Our model is the only to learn to condition on hand-drawn images and generate corresponding parametric CAD sketches, targeting a key component of an engineer's manual workflow. Such a model is also a prerequisite to the highly sought capability of "reverse engineering" observations of mechanical parts (e.g., scans, photos).
>
>
> *"the evaluation of the learned constraint graphs is fairly limited---it is essentially only demonstrated in one small figure."*
>
> The learned constraint graphs are evaluated both quantitatively and qualitatively in three locations:
> 1. Fig. 2 demonstrates the edit propagation that results from our inferred constraints.
> 2. Table 4 displays quantitative evaluation of the constraint model, including an ablation that demonstrates the criticality of training on noisy primitives.
> 3. Fig. 8 provides a qualitative demo of the ablation mentioned above. Generalization to noisy primitives is desired in application scenarios where, for example, the user has first drawn primitives in a haphazard way and now must constrain them, or where the primitive model has output imperfect coordinates.
>
>
> *"Willis et al. show how the standard auto-constrain feature in AutoCAD can be used to infer constraints for sketches produced by their model. Do the learned constrain graphs here address certain cases that cannot just be handled by this sort of simple post-processing?"*
>
> The auto-constrain feature of AutoCAD, while useful, requires manually setting the priority and tolerance for each individual constraint type. Instead, we propose a learned model that can adapt to the context of a given design automatically without the user toggling these settings.

---

> > ### Comment · Reviewer_DfnC · 2021-11-29
> > **Response**
> >
> > Thank you for the response. While I do still have some of the concerns in my initial review with respect to high-level takeaways, as the authors point out, the existing work with substantial overlap is concurrent as per ICLR guidelines, and so I raise my score.

---

### Official Review · Reviewer_ZmEf · 2021-11-02

**Correctness:** 4
**Technical Novelty And Significance:** 3
**Empirical Novelty And Significance:** Not applicable
**Recommendation:** 8
**Confidence:** 4

**Main Review:**

The SketchGraphs dataset has resulted in quite a few concurrent CAD sketch generation methods that roughly follow a similar approach, with a few variations. The proposed method differs from prior work SketchGraphs and Willis et al. in its generation approach, and its ability to generate explicit constraints, respectively, and from both prior and concurrent work (SketchGen, DeepCAD and Ganin et al.) in its ability to successfully infer a CAD sketches from hand-drawn images. I would consider this to be a good contribution, but unfortunately the authors do not compare their results to any of these methods, making it hard to judge the relative quality of the generated CAD sketches. Comparing the qualitative results manually across papers, does seem to show at least a similar level of quality as the concurrent work, and better quality than prior work, so I lean towards accept, conditioned on the authors including at least qualitative comparisons in their revision, and ideally quantitative comparisons as well.

Details (in order of importance):
- A comparison to prior work (SketchGraphs and ideally also Willis et al.) is needed. While both have a different generation approach than the proposed method, it should be possible to compare the constraint NLL and the full sketch NLL to SketchGraphs (as shown in Para et al. and Willis et al.), the primitive NLL to Willis et al., and the distributional statistics of generated sketches (possibly augmented with a few additional statistics to batter capture primitive and constraint quality) to both SketchGraphs and Willis et al. The qualitative results look quite good, so I would expect the quantitative results to be favorable as well. A comparison to concurrent work would be welcome, but is not necessary.
- The following concurrent work is missing and should be discussed briefly in the related work:
DeepCAD: A Deep Generative Network for Computer-Aided Design Models, Wu et al., ICCV 2021
- The exposition is generally very clear, I am just missing two pieces of information that I can't find mentioned explicitly:
--- Are types of primitives and constraints represented as separate 'type' tokens? If not, how are they represented? And if they are separate tokens, how are they represented in the constraint model, where the outputs are pointers rather than one-hot vectors?
--- In the standalone primitive model, each primitive is represented by multiple tokens, but constraints only refer to either full primitives or sub-primitives. How is a reference to a full primitive obtained if only token embeddings are available? The authors only mention briefly: "From the tuple of representations for each primitive, we extract reference embeddings for both the primitive as a whole (e.g., a line segment) and each of its sub-primitives (e.g., a line endpoint) that may be involved in constraints.", but the authors don't mention how this is done.
- For the primer-conditional generation, it might be good to discuss briefly if the primitives of an incomplete sketch need to be given in a similar order as during training (the typical order a designer would create the primitives in).
- Willis at al. and Ganin et al. are missing the venue in the bibliography.

**Summary Of The Paper:**

The authors present a generator for constrained CAD sketches that can also be used to convert hand-drawn line drawings to constrained CAD sketches or to auto-complete partial sketches. Two transformer models are used, the first model generates CAD primitives, optionally conditioned on a line drawing image or on a partial CAD sketch, and the second model generates constraints conditioned on the generated primitives. The authors show good qualitative results and quantitatively improved CAD generation performance over two simple baselines.

**Summary Of The Review:**

The proposed methods contributes a new generation approach and explicit generation of both primitives and constraints over prior methods; and reconstruction of CAD sketches from hand-drawn images over both prior and concurrent work. The authors also show good qualitative results. However, as a big minus, the result are not compared to any related work. On the balance though, I am still leaning slightly towards acceptance, since the contributions are good, and the qualitative results are convincing enough.

---

> ### Author Response · Authors · 2021-11-23
> **Response to Reviewer ZmEf**
>
> Thanks, Reviewer ZmEf, for your thoughtful comments. We address them below.
>
> *Reference to DeepCAD:*
>
> Thanks for pointing out the missing reference to Wu et al., 2021. In the revision, we have updated the related work section accordingly.
>
>
> *"Are types of primitives and constraints represented as separate 'type' tokens? If not, how are they represented? And if they are separate tokens, how are they represented in the constraint model, where the outputs are pointers rather than one-hot vectors?"*
>
> Primitive and constraint types are represented using a pre-specified range of each model's value tokens. For example, for primitives, as described in the Tokenization paragraph of Sec. 3, a small set of the value tokens (four tokens for Arc, Line, Circle, and Point) are reserved for type specification. Three value tokens are also reserved for Start, Stop, and Pad.
>
> To clarify, the constraint model's output consists of both categorical types and references to primitives. At any given step during inference, the output dot product is computed between the decoder's output representation and the concatenation of the learned embeddings for the constraint types and the (sub-) primitive encodings. That is, the softmax output layer of the constraint model selects from the union of both categoricals (for the constraint types) and the references. We may view this still as producing a pointer, but the possible targets for the pointer are not only the primitive references, but the set of constraint types as well.
>
>
>
> *"How is a reference to a full primitive obtained if only token embeddings are available?"*
>
> We took a simple approach where, for each primitive type, a subset of the tuple of learned encodings (as output by the primitive model) are tasked with representing either the entire primitive or a specific sub-primitive. For example, for a line, which consists of a tuple of length 5 (type Line, x0, y0, x1, y1), we use the encoding at index 0 to represent the line segment as a whole, index 1 (the x coordinate of the first endpoint) to represent the first endpoint, and index 3 (the x coordinate of the second endpoint) to represent the second endpoint. We note that, after several layers of transformations in the primitive model, the representations at these indices do not solely represent the original token but instead learn to incorporate the information needed to represent the (sub-) primitive they were assigned. Note that this mapping is arbitrary and there are other design routes that could produce equivalent results. See Appendix C for the reference index map for each primitive type.
>
>
>
>
> *"For the primer-conditional generation, it might be good to discuss briefly if the primitives of an incomplete sketch need to be given in a similar order as during training (the typical order a designer would create the primitives in)."*
>
> We do observe empirically that the dataset sequences adhere to certain patterns in the ordering of primitives, and thus the model performs better at test time when examples come from this same distribution. In addition to the random permutation ablation (at training time) shown in Table 2, we have added an additional experiment where we take the regular unconditional primitive model (trained on the dataset's orderings) and test it on randomly permuted sequences. As now described in section A.2 and Table 5, the NLL increases to 104.7 bits/sketch, compared to 66.6 bits/sketch when testing on the original test set. This aligns with what we observe empirically, where sketch completions are more likely to degrade if the prefix is shuffled. This effect can be observed, for example, with rectangles, where the model's performance suffers if the line segment ordering does not adhere to the default ordering from Onshape's rectangle tool.
>
>
> *"Willis at al. and Ganin et al. are missing the venue in the bibliography."*
>
> Thanks for pointing this out. We've gone ahead and added the venues for these two as well as for Para et al. To our knowledge, the NeurIPS acceptances were not publicly announced until after the ICLR deadline.

---

> > ### Comment · Reviewer_ZmEf · 2021-11-27
> > **The added comparisons and clarifications raise my assessment of the paper.**
> >
> > Thanks for pointing out that Willis et al. is indeed considered concurrent work. The added comparison to SketchGraphs shows a relatively clear quantitative advantage for the proposed method, although for the qualitative comparisons, providing a more direct side-by-side comparison in the paper rather than referring to figures in the other papers would be helpful. After reading the other reviews and considering the added comparisons and clarifications provided by the authors, I am raising my score by by 2 points.

---

### Author Response · Authors · 2021-11-23
**General Response to All Reviewers**

Thank you, reviewers, for your thoughtful comments. Here we address common points among the reviews and describe the resultant updates reflected in the revision. In addition, we will respond to each reviewer separately regarding points particular to their review.

**Comparison to related work:**
We have added both a quantitative and qualitative comparison to Seff et al., 2020 (Table 4, Appendix G, and Appendix H) and a qualitative comparison to Willis et al., 2021 (Appendix H).

Note that the generative model from Seff et al. does not output any numerical parameters for the primitives; it relies solely on the constraint graph to determine primitive placement. We make our best attempt to adapt their model to our setting to enable a direct comparison with respect to negative log-likelihood (NLL). Moreover, there is no code available for Willis et al.; direct quantitative comparison is not possible as both the filtered dataset and output space differ between our models and theirs. Comparing models with different output semantics is notoriously difficult in the context of NLL.

**Concurrent vs. prior work:**
At the risk of being pedantic, we feel it is useful to address the question of concurrent vs. prior work. The ML community has grown significantly over the past several years, and there is a variation among opinions regarding what constitutes "concurrent" vs. "prior" work. Additionally, since individual conferences often subscribe to different positions in this respect, we feel it is important to highlight here the precise language used within the ICLR 2022 Reviewer Guide:

"We consider papers contemporaneous if they are published (available in online proceedings) within the last four months. That means, since our full paper deadline is October 5, if a paper was published (i.e., at a peer-reviewed venue) on or after June 5, 2021, authors are not required to compare their own work to that paper."
See https://iclr.cc/Conferences/2022/ReviewerGuide for the full text.

To our knowledge, the only paper on CAD sketch generation meeting this definition of prior work is Seff et al., 2020, which was published at ICML 2020 Workshop on Object-Oriented Learning. As stated in our paper, a direct comparison is complicated by the fact that they do not model the task in its full generality (no primitive parameters). Nevertheless, in our revision, we have adapted their method in order to provide a quantitative comparison to a stronger baseline.

The following works, all of which we reference in our paper, are concurrent to our work according to ICLR's definition:
*    Willis et al., 2021: presented at CVPR 2021 SketchDL Workshop (June 19, 2021)
*    Wu et al., 2021: presented at ICCV 2021 (October 11–17, 2021)
*    Para et al., 2021: to appear at NeurIPS 2021 (December 6–14, 2021)
*    Ganin et al., 2021: to appear at NeurIPS 2021 (December 6–14, 2021)

Not only does the concurrency of these works meet ICLR's requirements, but we note that a version of our paper (with completed methods and experiments) was publicly presented in June 2021 (we do not specify details due to the anonymization rules).


**Additional updates:**
Based on valuable feedback from the reviewers, we have made the following additional updates to the paper:
*    An additional ablation consisting of testing the standard unconditional primitive model on randomly permuted sequences (Table 5, Appendix A.2).
*    Additional system details and figures illustrating the tokenization procedure (Appendix C).
*    A visual demonstration of quantizing SketchGraphs sketches at various bit widths (Appendix D).

---

### Decision · Program_Chairs · 2022-01-20

**Decision:**

Accept (Poster)

**Comment:**

The paper describes an approach for automatically generating CAD sketches, including both the primitives that describe the drawing, as well as the constraints that describe relationships between the primitives that need to be maintained even if the primitives are changed. This is an important problem that is starting to receive a lot of attention from the literature.

Overall, the paper is very well executed and the results are quite compelling.

There were some concerns about the relationship with the work by Willis et al. and other papers that were published around the time when this paper was submitted. There is still some novelty in this paper relative to those works as argued in appendix H, but it would have been really good to have a more quantitative comparison. However, the authors pointed out that this work was concurrent as opposed to prior work as per the ICLR reviewer guidelines.

Overall, given the quality of this paper and the guidance given in the ICLR reviewer guide, most reviewers agree with the meta-reviewer that this paper should be accepted (the lowest reviewer still indicated it is above the acceptance threshold). However, there is some discomfort around not having an explicit comparison with very closely related work that ultimately was published before this paper.